



# Performance and polarization response of Slit Homogenizers for the GeoCarb Mission

Sean Crowell[1], Tobias Haist[2], Michael Tscherpel[2], Jérôme Caron[3], Eric Burgh[4], and Berrien Moore III[1]

[1]University of Oklahoma, Norman, Oklahoma, USA
[2]Institut für Technische Optik, University of Stuttgart, Germany
[3]TNO, Optics Department, Stieltjesweg 1, 2628 CK Delft, The Netherlands
[4]Lockheed Martin Advanced Technology Center, Palo Alto, California, USA

**Correspondence:** Sean Crowell (scrowell@ou.edu)

**Abstract.** The observing strategy of the Geostationary Carbon Observatory (GeoCarb), which is a "step and stare" approach, can lead to distortions in the instrument spectral response function (ISRF) when there are gradients in brightness across instrument field of view. These distortions induce errors in the retrieved trace gases. In order to minimize these errors, the GeoCarb instrument design was modified to include a "slit homogenizer" whose purpose is to scramble the pattern of the incoming light and effectively remove the ISRF distortions causing by the variations in illumination across the slit. As a risk reduction, Geo-

Carb procured six different homogenizers and had them tested for performance in a bench-top optical system. The major finding is that the homogenizer performance depends strongly on the polarization of the incoming light, with the sensitivity growing as a function of wavelength. The width of the ISRF is substantially smaller when the light is vertically polarized (orthogonal to the slit length) compared to horizontally polarized (parallel to the slit length), and the throughput is accordingly reduced.

These effects are due to the effects of the gold coating and high incidence angles present in the GeoCarb homogenizer design, which was verified using a polarization-dependent model generalized from previous homogenizer modeling work. The results strongly recommend controlling the polarization of the light entering a similar implementation for other instruments attempting to mitigate scene illumination non-uniformity effects, as well as a robust characterization of the polarization sensitivity of all key subsystems.

## 15 1 Introduction

The Geostationary Carbon Observatory (GeoCarb; Moore III et al. (2018)) will launch in the next few years and make observations of carbon gases with the goal of understanding the carbon cycle of the Americas. The tight tolerances on trace gas retrieval accuracy have placed a very strong constraint on the instrument design (e.g., the requirements in Table 1) to be able to meet the goals of the mission (Moore III et al., 2018). The instrument design has been optimized to maximize signal-to-noise

ratio (SNR) as well as minimizing stray light, temporal sampling latency, and the spatial sample size. However, previous studies have shown that even for perfect instruments (i.e., no stray light, high SNR), retrievals of trace gases from observed radiances in the near infrared (NIR) and shortwave infrared (SWIR) bands may be biased by different error sources such as clouds and



aerosols, surface properties, and calibration errors (Butz et al., 2012; Connor et al., 2016; O'Dell et al., 2018; Kiel et al., 2019).
Simulation studies for the TROPospheric Monitoring Instrument (TROPOMI) (Landgraf et al., 2016; Hu et al., 2016) have also

shown that variations in brightness within a scene, which affect the shape of the instrument spectral response function (ISRF),
can lead to errors in NIR and SWIR trace gas retrievals. Previous studies in support of other missions suggested the use of a
"slit homogenizer" (SH) could mitigate these errors successfully.

With these considerations in mind, SH test articles with three different depths were procured from two vendors for performance
testing. In parallel, existing models (Bauer et al., 2017; Caron et al., 2019) were extended to analyze the results. These studies

were done in advance of modifying the GeoCarb instrument design by replacing the traditional "air slit" with a 1-D SH. The
goals of the study were 1) to determine whether the 1-D SH reduces the impacts of scene inhomogeneity, and 2) to determine
which device has the best performance. The design and results of the study are reported herein.

The paper is structured as follows. Section 2 gives background on the issue of scene inhomogeneity and the SH as a potential
solution. Section 3 describes the modeling approach. Section 4 presents the performance measurement approach. Section 5

gives the experimental results and comparisons with models. Section 6 and Section 7 provide a summary discussion and
conclusions and suggestions for future work.

## 2  Background

### 2.1  GeoCarb

The Geostationary Carbon Cycle Observatory (GeoCarb) was selected as the recipient of NASA's second Earth Venture Mission

(EVM-2) in 2016, and will make daily measurements of total column carbon dioxide (XCO$_2$), methane (XCH$_4$), and carbon
monoxide (XCO) from geostationary orbit, nominally at 85°W. The ability to make daily scans of the western hemisphere will
enable a "step-change" in our understanding of the carbon cycle (Moore III et al., 2018). From geostationary orbit over the
Americas, GeoCarb will observe reflected sunlight in the 0.76$\mu$m (O$_2$-A), 1.61$\mu$m (Weak CO$_2$), 2.06$\mu$m (Strong CO$_2$), and
the 2.32$\mu$m (CO/CH$_4$) spectral regions. Additionally, Fraunhofer lines in the O$_2$-A band allow for retrieval of Solar-Induced

Fluorescence (SIF) (Frankenberg et al., 2012). GeoCarb will use a 4-channel long slit spectrograph with a field of view of 4.3
degrees by 30 arcseconds, corresponding, at nadir, to about 2700 km in the North-South (along slit) direction and 5.4 km in the
East-West (across-slit) direction. Using a scan mirror system comprising two orthogonal flat mirrors, the slit can be scanned
across the Earth's surface. Each element of a scan consists of about 1000 N-S spatial samples with an integration time of 8.6s
taken at a scan cadence of 9.4 seconds for adjacent slit pointings, including time for movement and settling at a new pointing.

A single day of observations will yield about 4 million soundings, each with a spatial field-of-view of roughly 2.7 km (N-S)
by 5.4 km (E-W) at the sub-satellite point. Additional instrument specifications are given in Table 1.





|  | O2A | WCO2 | SCO2 | CH4/CO |
|---|---|---|---|---|
| Center WL ($\mu$m) | 0.764 | 1.61 | 2.06 | 2.32 |
| WL Range ($\mu$m) | 0.757-0.771 | 1.592-1.621 | 2.045-2.085 | 2.301-2.346 |
| Resolving Power | >16118 | >15941 | >15147 | >15163 |
| Reference* SNR | >395 | >389 | >302 | >254 |

**Table 1.** GeoCarb Instrument Requirements. The top row gives the designation for the four GeoCarb spectral bands: the oxygen A-band (O2A), the weak $CO_2$ band (WCO2), the strong $CO_2$ band (SCO2), and the longest wavelength band with absorption lines for $CH_4$ and CO (CH4/CO). * Reference SNR refers to the signal-to-noise ratio for the representative conditions at Railroad Valley, Nevada at local noon on the summer solstice.

## 2.2 Scene Inhomogeneity and Slit Homogenizers

Following (Caron et al., 2019), the ISRF is defined by

$$ISRF(\lambda - \lambda_o) = \left[ S\left(\frac{x}{M}\right) * PSF_{spec}(x) * PSF_{det}(x) \right] * \delta\left( x - \frac{\lambda - \lambda_o}{k} \right) \tag{1}$$

where

- $S$ is the scene illumination and $M$ is the spectrometer magnification factor that maps the slit width dimension to the image on the detector,

- $PSF_{spec}$ and $PSF_{det}$ represent the 1-D optical response (including aberrations and diffraction) of the spectrometer and the response of the detector pixel (treated as a boxcar function), respectively, after averaging the corresponding 2-D function in the along slit direction, and

- $\delta$ represents the change of variable from the detector position $x$ (where $x = 0$ is the center of the detector) to the wavelength grid centered at wavelength $\lambda_o$, assuming spectral dispersion $k$.

In trace gas retrievals, high-resolution radiance spectra are convolved with the assumed ISRF, typically measured during preflight instrument characterization (e.g., Lee et al. (2017)), prior to comparison with observed radiances. Any mischaracterization of the ISRF can thus lead to errors in retrieved trace gases. One such mischaracterization occurs when brightness varies across width of the slit field of view (FOV), so that the function $S$ is not constant. In this case, the ISRF is distorted relative to the ISRF measured pre-flight. These brightness variations can be due to gradients in surface albedo caused by different land surface types within the scene as well as partial cloud cover. In Landgraf et al. (2016) and Hu et al. (2016), these sorts of variations were shown to lead to significant errors in $XCH_4$ and XCO using simulations. Figure 1 shows one example of MODIS-observed albedo variations in three GeoCarb-relevant bands and the resultant impact on the normalized ISRF at the center wavelength in each of the four GeoCarb bands. The left panel shows the variation in $500\,\mathrm{m}$ MODIS black sky albedo within a single GeoCarb footprint for the three MODIS bands nearest the GeoCarb spectral bands: Band 2 ($0.85\,\mu$m), Band





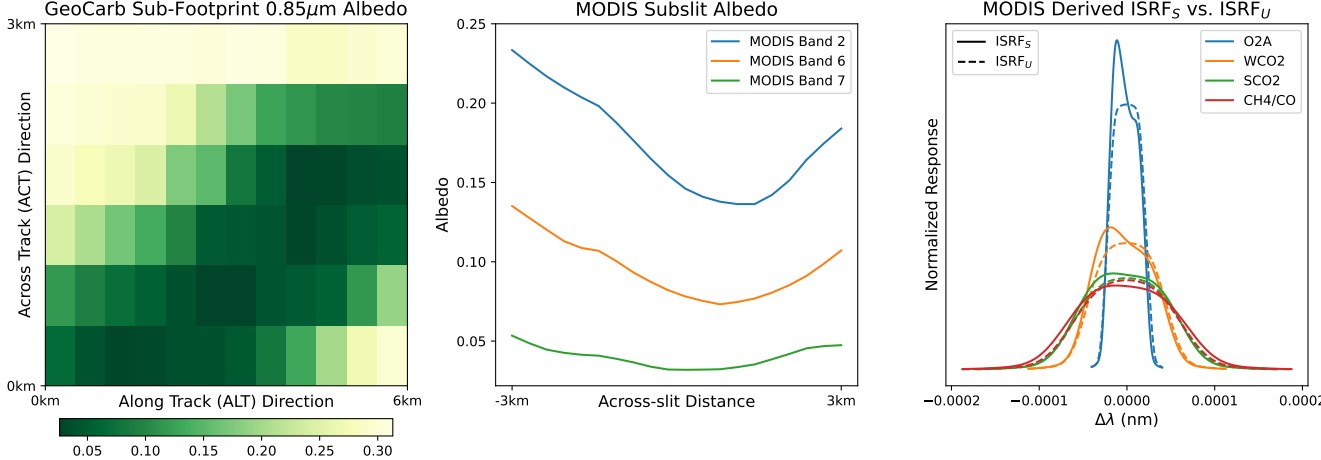

**Figure 1.** Variation in MODIS Band 2 ($0.85\,\mu$m) observed 500m black sky albedo for a single GeoCarb footprint (left); Band 2, Band 6 ($1.64\,\mu$m), and Band 7 ($2.1\,\mu$m) albedos averaged in the N-S direction (center); and the resultant normalized ISRFs for the center wavelength in each of the four GeoCarb bands in Table 1 (right). $ISRF_S$ denotes the ISRF computed from (1) with the illumination profile in the center panel, while $ISRF_U$ uses a flat illumination profile. In this figure, the Band 7 albedo was used for the CH4/CO band for simplicity.

6 ($1.64\,\mu$m), and Band 7 ($2.1\,\mu$m). The right panel shows the effect of a strong transition from brighter illumination to darker illumination: the resultant functions $ISRF_S$ are skewed to the left relative to the uniform illumination derived functions $ISRF_u$.

1-D SHs have been discussed previously (Sierk et al., 2017; Bauer et al., 2017; Caron et al., 2019) as a method of reducing the influence of brightness variations on the ISRF. Successful reduction of inhomogeneities was demonstrated at the breadboard level (Sierk et al., 2017) and was thus incorporated into the Sentinel 5 design (Bauer et al., 2017). Based on the experimental and theoretical results in these previous studies, prototype $36\,\mu$m wide 1-D SH devices were procured from two vendors with depths of approximately $0.5\,$mm, $1.0\,$mm, and $1.5\,$mm, yielding six total devices. Each SH device consists of a thick slit made

with two parallel mirrors, as is depicted in Figure 2. The distance between the two mirrors is equivalent to the slit width. The two mirrors will create multiple reflections that homogenize the radiance at SH input. After a certain distance equal to the SH depth the illumination reaches the SH output and is homogenized. It is then delivered to the spectrometer. The performance testing of these devices and the theoretical interpretation of the results are the subject of this work.





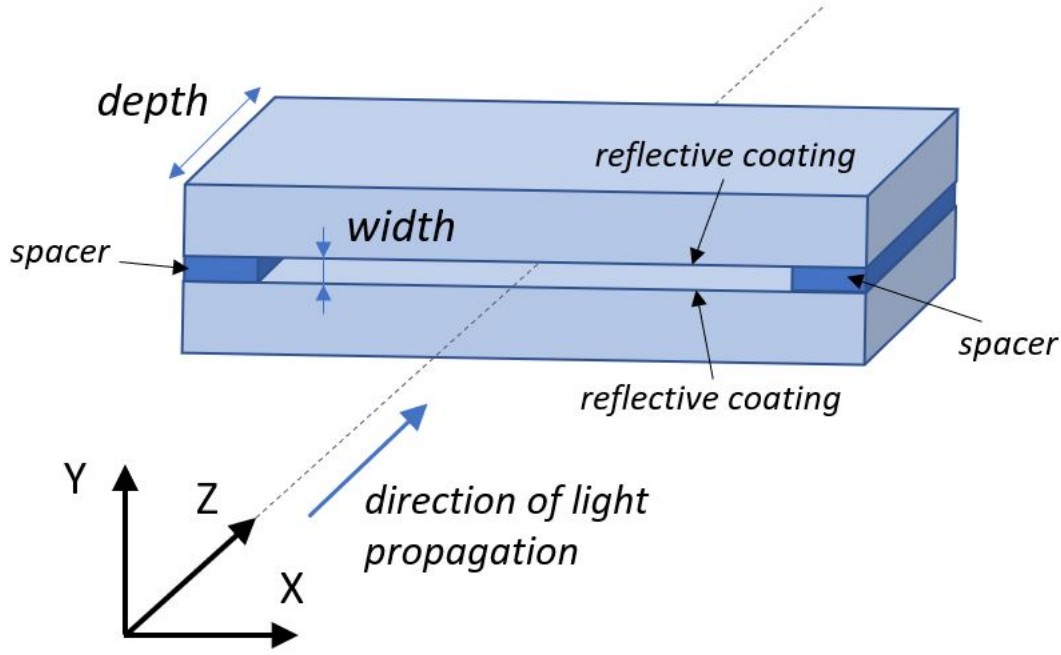

**Figure 2.** The slit homogenizer is composed of two parallel mirrors separated by spacers. The slit opening is defined by its width and length as for a regular slit. The slit homogenizing function comes from the third dimension (depth) where multiple reflections homogenize the illumination.

## 3 Slit homogenizer response modelling

### 3.1 Principles of slit homogenizer modelling

The simulated GeoCarb 1-D SHs are defined by two parameters: the slit width $w$ and the slit homogenizer depth $L$ as depicted in Figure 2. $L$ is the extent of the reflective surfaces in the direction of light propagation and defines how many multiple reflections take place.

Modelling the SH response was discussed in detail in other references (Caron et al., 2019; Bauer et al., 2017) and so we will
summarize the main aspects here. The model computes the intensity distribution at the SH output for every possible position of a stimulus (source point, or diffraction PSF) at the input plane. The obtained intensity profiles are stored in a "transfer matrix" (Caron et al., 2019). The response of the SH to an arbitrary intensity distribution is then computed by proper weighting and addition of the relevant transfer matrix columns.

Let us assume that a point source is imaged on the SH entrance. When the SH entrance is observed from the SH output it
appears to be illuminated with multiple point sources which are aligned and placed at regular intervals: the original point source as well as all its mirror images after reflection on the two mirrors as is shown in Figure 3. To compute the intensity



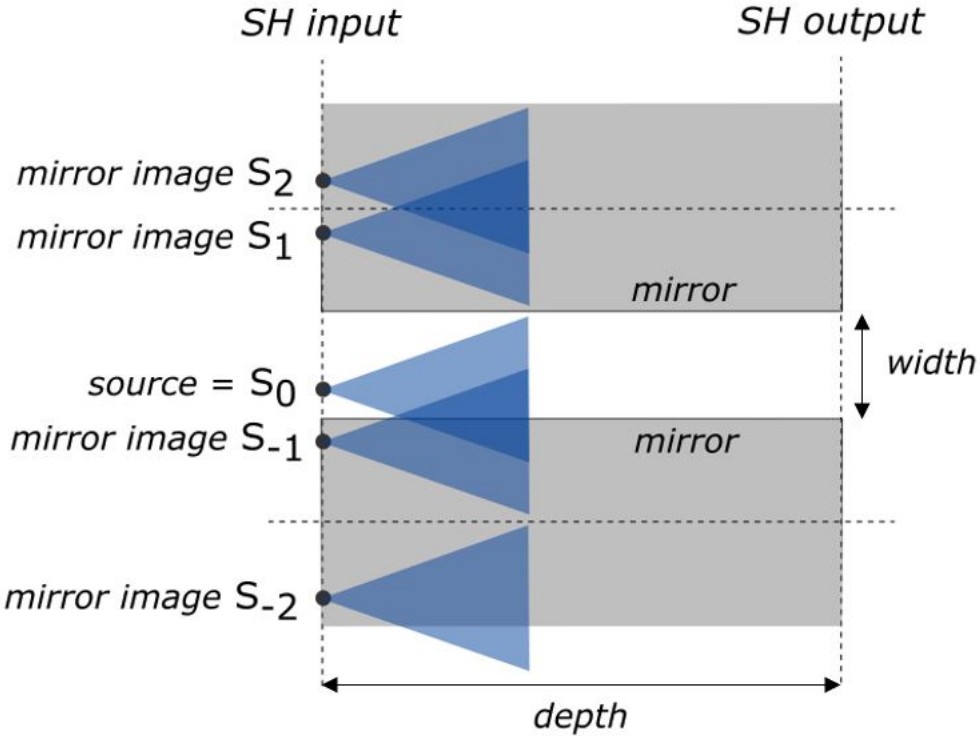

**Figure 3.** A point source S0 is imaged at the SH input. The rays emitted by S0 are reflected multiple times by the two mirrors, creating images of S0 (noted $S_1$, $S_2$, …, etc and $S_{-1}$, $S_{-2}$, …, etc). The total illumination seen at the SH output is the superposition of the rays emitted by the original point source S0 and all its mirror images.

at the SH output one needs to add the contributions from the original point source plus all its mirror images, accounting for possible interferences due to their optical path difference.

This can be done in two ways:

– The interference model uses simple complex addition. The input stimuli are source points emitting light within a cone that corresponds to the system F-number. This type of model is quite flexible and fast but gives only approximate results, and might not exactly conserve energy (Caron et al., 2019).

– The diffraction model expresses the input stimuli as a diffraction point spread function (PSF) obtained with the optical system placed in front of the SH input plane, and a diffraction integral is used for propagation through the SH. This

type of model is more accurate but its computation time is significantly longer. For a rectangular pupil, thanks to the separability of the along slit (X in Figure 2) and across slit (Y in Figure 2) directions a 1-D diffraction model can be used, with an acceptable computing time. Modelling a circular pupil like GeoCarb's is more demanding and was not performed.





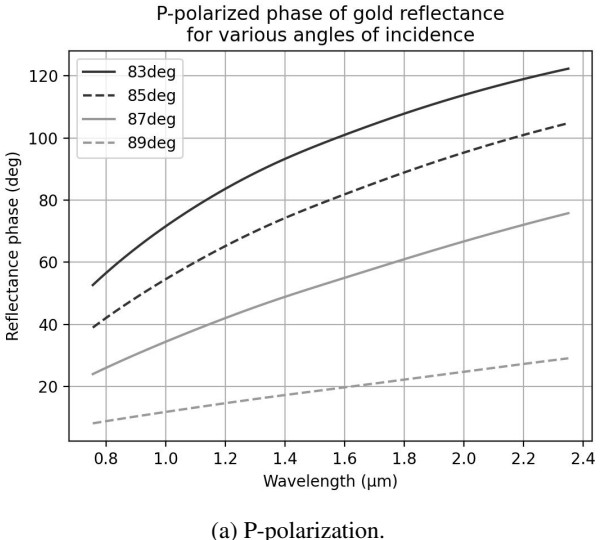

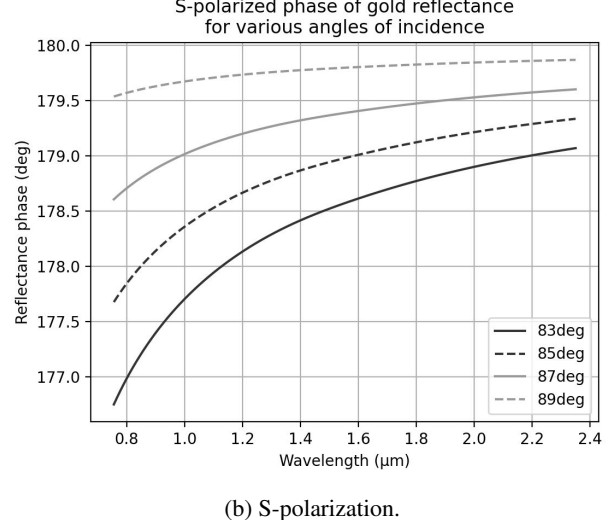

(a) P-polarization.

(b) S-polarization.

**Figure 4.** Phase shift at reflection on a thick gold layer, computed from Rakic gold index parametrization (Rakić et al., 1998) with definitions of polarization axes from Macleod (2017). (a) P-polarization, (b) S-polarization. The 180 ° shift in P-pol discussed in Figure 5 is not included.

In the experiments that follow, we utilize the diffraction integral technique for maximum accuracy.

## 3.2 Specifications due to gold coating

In this work, existing SH models (Bauer et al., 2017) were upgraded to include the impacts of the properties of gold reflective coatings, particularly with regard to polarization, which were not included in the previous studies. These upgrades were undertaken in large part to explain the behavior observed in the measurements described later in the paper that seem to result from GeoCarb's narrow slit and relatively fast beam (f/4.5). The gold optical constants were computed with a Lorentz-Drude model with parameters taken from Rakić et al. (1998). The phase of the mirror complex reflectance plays a crucial role in the SH response, so we now discuss it in detail. The definition of the phase shift that occurs at reflection depends on two aspects:

- The sign of the phase is changed depending on the complex notation that is used, with positive or negative complex exponentials $exp(+i\omega t)$ or $exp(-i\omega t)$

- If, at normal incidence, the vectors for incident and reflected electric fields are equal or differ by a minus sign, a 180° phase shift can be induced in the reflectance phase.

Our equations to compute the gold reflectance are taken from Macleod (2017). A positive complex exponential $exp(+i\omega t)$ is used, and the electric fields (incident and reflected) coincide at normal incidence for both S and P polarization. The computed phase shifts are displayed on Figure 4.





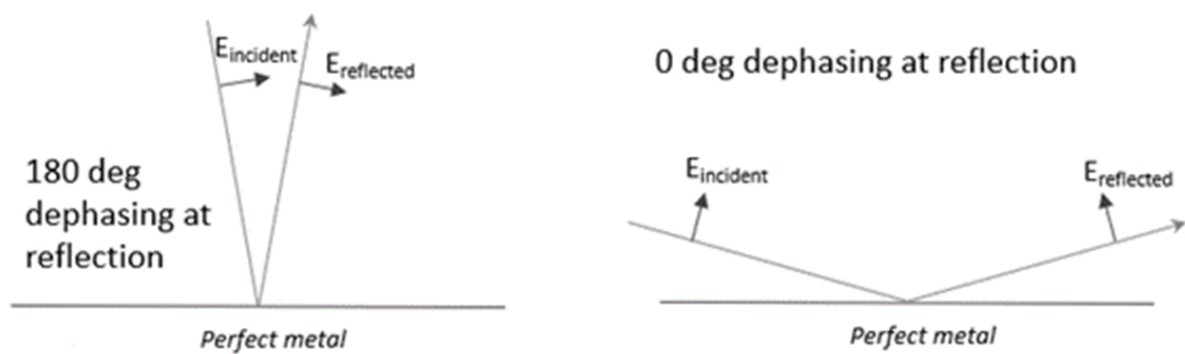

**Figure 5.** Definition of phase reflectance in P-polarization. (left) situation close to normal incidence. The P-polarized electric fields before and after reflection are defined according to McLeod (2017) and are pointing in the same direction. For an ideal metal with infinite conductivity the phase shift at reflection is 180 deg. (right) situation close to grazing incidence. We adopt the opposite definition as McLeod (2017) with electric fields pointing in opposite direction if the incidence angle was reduced to zero. At grazing incidence it means the electric fields are pointing towards the same direction. For an ideal metal the phase shift at reflection is now 0 deg due to the different definition.

In order to incorporate the complex reflectance in the SH model, as discussed in Caron et al. (2019) it is necessary to make a
180° correction for the P-polarized phase. A simple example helps to explain the need for this additional phase shift. Consider reflection on a substrate made of an ideal metal (with infinite conductivity). In the convention of Macleod (2017), where the incident and reflected electric field coincide at normal incidence, the phase shifts obtained at reflection are equal to 180° at all angles and for both polarizations. These 180° dephasings are an intrinsic property of the ideal metal.

When the angle of incidence is increased to large values, the angle between incident and reflected rays increases by 180° with
respect to the situation at normal incidence (Figure 5). In this process, the P-polarized electric field has also rotated by 180°. The phase shift of 180° between incident and reflected electric fields in P polarization pointing in the same direction at normal incidence, becomes a phase shift of 180° between two electric fields pointing now in the opposite direction at grazing incidence due to the beam rotation. This phase shift is equivalent to a 0° phase shift for 2 electric fields that are pointing in the same direction at grazing incidence (Figure 5, right). In the SH model, we will assume that the direction of the electric fields does
not change at reflection at grazing incidence. It is therefore necessary to add an extra phase offset of 180° in P-polarization. No such offset is required in S-polarization.

### 3.3 Sizing of the GeoCarb SH Prototypes

An earlier version of the SH simulator (without polarization dependence) detailed in Caron et al. (2019) was used to estimate the optimal depth of the homogenizer for all four GeoCarb bands, which share a single slit in the GeoCarb design. As is
discussed above, the simulator generates a transfer matrix that depends on depth and wavelength.





If only pure geometric optics is considered and if the instrument has a rectangular pupil, it can be easily shown that after a propagation distance equal to multiples of $2 * F\# * w$ the beam uniformly illuminates the slit output. From the perspective of geometric optics, if the SH depth is equal to a multiple of $2 * F\# * w$ the slit homogenization is then perfect for a rectangular pupil. In reality this simple rule of thumb is not valid due to the presence of interferences.

145 To determine the optimum depth for the slit homogenizer for GeoCarb, we simulated polarization-independent transfer functions for a range of depths at all four wavelengths of interest (listed in Table 1) using the previously documented SH model (Bauer et al., 2017). We then randomly generated an ensemble of input scenes with inhomogeneities that had an average coefficient of variation of scene brightness of 20% on size scales of about 1/5th of a slit width. We applied the computed transfer function for each depth and the resultant coefficient of variation measured. We then calculated the ratio of the coefficient of variation of the output to the coefficient of variation on the input, dubbed the "reduction factor", and then calculated the median of the ensemble's reduction factors for each wavelength at each slit depth.

After an initial ramp up in performance at the lowest depths, the reduction factors were relatively independent with homogenizer depth; however, there were conspicuous dips in performance at specific depths that were not coincident for all four wavelengths. These dips are due to some periodicity in the interference patterns as the light propagates along the depth of the slit. GeoCarb uses a single $36\mu$m wide slit, so we chose to investigate three depths that optimized performance across all four bands simultaneously, which were approximately 0.5mm, 1mm, and 1.5mm.

## 4 Experimental characterization

In order to assess the performance of the different six SH prototypes, a breadboard test setup was constructed at the Institut für Technische Optik (ITO). This section details the optical design and fabrication employed in the testing.

### 4.1 Measurement setup

The homogenization performance was evaluated based on recordings of the near-field intensity distribution at the exit of each SH while the input was illuminated with different illumination patterns (realized by a knife-edge). All measurements were performed at four different wavelengths ($0.76\,\mu$m, $1.62\,\mu$m, $2.05\,\mu$m and $2.33\,\mu$m) and for polarizations parallel (across track, ACT) as well as orthogonal (along track, ALT) to the slit. Additionally, the throughput efficiency was measured.

Figure 6 depicts the setup used for the measurements. To achieve spatially incoherent light we employed a diffuser approach based on the combination of a fixed and a rotating diffuser (ground glass, 220 grit from Edmund Optics). Fourier geometry (L1 in Figure 6, $f_1$=100 mm, image-sided telecentric) is used to homogenize the light in the plane of the knife-edge and to make sure that every point in this plane receives the same angular light distribution (telecentric illumination).

The following mono-mode fiber coupled monochromatic sources have been employed:

– $0.76\,\mu$m: tunable Laser from Sacher Lasertechnik Group, model TEC 500, serial no. P-770-0517-01003





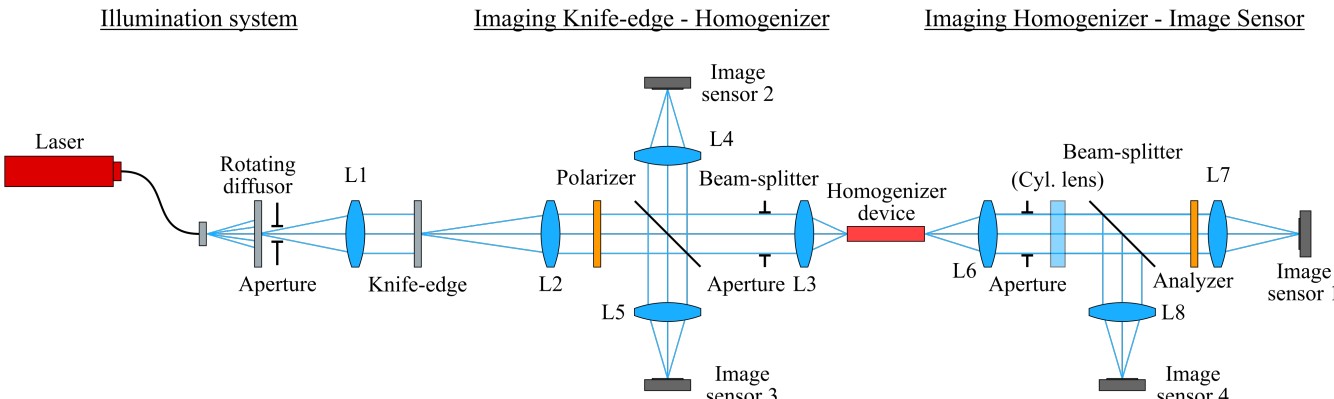

**Figure 6.** Setup for the near field measurements.

- $1.62\,\mu m$: tunable Laser from Sacher Lasertechnik Group, model TEC 500, serial no. P-1650-0615-00941

- $2.05\,\mu m$: YE3085 single-mode fiber pigtailed diode laser from NASA Jet Propulsion Laboratory at approximately $3\,mW$.

- $2.33\,\mu m$: EP2327-0-DM-TP39-01 from Eblana Photonics at approximately $2.6\,mW$.

The knife edge (Edmund optics # 36-137) with variable orientation (micromechanical adjustable mount) serves to obscure part
of the slit and thus introduce a sharp cutoff in illumination across the homogenizer, simulating an extreme version of the typical
image taken from space by the GeoCarb instrument.

The knife edge is imaged by a double-sided telecentric system at an F-number of 4.5 into the entrance plane of the SH. For
$0.76\,\mu m$ and $1.62\,\mu m$ we used two achromatic lenses (Thorlabs AC254-150-B, focal length $f_2 = 150\,mm$ and Thorlabs AC050-
010-B-ML, focal length $f_3 = 10\,mm$). The paraxial magnification is -0,067. For $2.05\,\mu m$ and $2.33\,\mu m$ we used an air-spaced
achromat (Thorlabs ACA254-200-D, focal length $f_2 = 200\,mm$) and a plano-convex spherical lens (Thorlabs LA0309-E, focal
length $f_3 = 15\,mm$). The paraxial magnification is -0.074. Both systems were optimized in Zemax to minimize the RMS spot
radius under the requirement to maintain telecentricity.

Coherence reduction and homogeneity were verified by measurement of the speckle contrast in the plane of the SH. To validate
the image quality and homogeneity at $0.76\,\mu m$, we placed an image sensor (Ximea MQ042RG-CM) without cover glass at the
entrance plane of the homogenizer (see Figure 7). The line cut along the image of the knife edge shows a standard deviation of
$1\,\%$ after subtracting the linear contribution. We took the contrast $(I_{max} - I_{min})/(I_{max} + I_{min})$ as a measure of homogeneity.
For $2.05\,\mu m$ and $2.33\,\mu m$ this results in $0.36\,\%$ and $0.40\,\%$ (after applying a Gaussian filter with $\sigma = 15\,pixels$ to reduce noise).





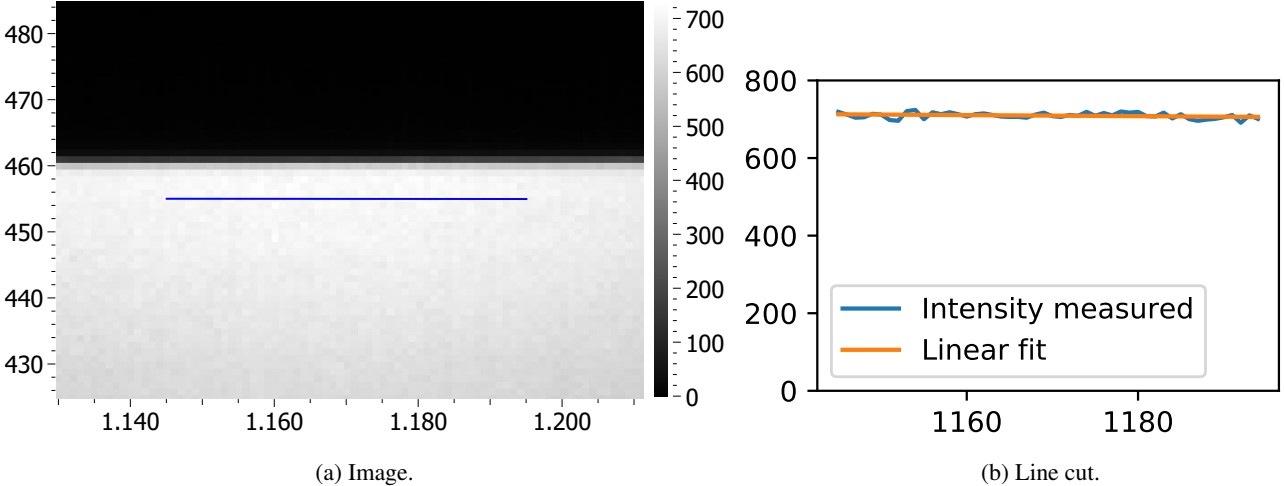

(a) Image.                                (b) Line cut.

**Figure 7.** a) Image of the pickoff mirror in the position of the device input, taken by image sensor without cover glass. b) intensity along the horizontal line depicted in a)

The slit homogenizer can be moved in all 3 dimensions using high precision piezo stages (Physik Instrumente PI Q-545.240 linear stage, PI E-873.3QTU controller, repeatability error of $200\,\mathrm{nm}$). The orientation of the SH mount is parallel to the optical

axis to better than $\pm\,0.1°$ (aligned using two additional Helium-Neon alignment lasers).

The intensity distribution at the exit of the SH is imaged by a double-sided telecentric system (L6 and L7 in Fig. Figure 6) onto the image sensor with an F-number of 4.0.

For the measurements at $\lambda = 0.76\,\mathrm{\mu m}$ and $1.62\,\mathrm{\mu m}$ L6 and L7 are achromatic lenses with focal lengths of $f_6 = 10\,\mathrm{mm}$ and $f_7 = 200\,\mathrm{mm}$ (Thorlabs AC050-010-B-ML, AC254-200-B-ML). The simulated paraxial magnification of this system is -20 (ALT)

and -25.9 (ACT) respectively.

In order to be able to judge homogenization performance and to see possible crosstalk it is preferable to use anamorphotic imaging so that vertical structures in the output plane of the device are resolved while still the lateral position along the slit on the input side is maintained on the image sensor side. To this end we used an additional cylindrical lens ($\lambda = 0.76\,\mathrm{\mu m}$: LK1336RM-B, $\lambda = 1.62\,\mathrm{\mu m}$: LK1336RM-C) while imaging the exit of the device onto the image sensor.

For $\lambda = 2.05\,\mathrm{\mu m}$ and $\lambda = 2.33\,\mathrm{\mu m}$, it was not possible to achieve a near diffraction-limited anamorphotic setup with stock lenses. Instead, a rotationally symmetric system was used. This way only the device output plane was imaged sharply onto the image sensor, but the measurements at $\lambda = 0.76\,\mathrm{\mu m}$ and $\lambda = 1.62\,\mathrm{\mu m}$ suggested that this does not result in a loss of information. L6 and L7 are changed to LA0309-E (focal length $f_6 = 15\,\mathrm{mm}$) and ACA254-300-D (focal length $f_7 = 300\,\mathrm{mm}$). The (simulated) paraxial magnification of the system is $-31.8$ ($2.05\,\mathrm{\mu m}$) and $-32.7$ ($2.33\,\mathrm{\mu m}$).





The polarizer ($\lambda$ =0.76 µm: Thorlabs LPNIRE100-B, $\lambda$ =1.62 µm: LPIREA100-C, $\lambda$ =2.05 µm and $\lambda$ =2.33 µm: LPNIRA050-MP2) is placed between L2 and L3, together with a half-wave plate to rotate the polarization from the laser output.

In the NIR, a scientific CMOS image sensor (PCO edge 3.1 2048 x 2048 pixels, 6.5 µm pixel pitch) was used. At $\lambda$ =1.62 µm a Raptor Photonic InGaAs image sensor (Ninox 640 VIS-SWIR, 640 x 512 pixels, 15 µm pixel pitch) and at longer wavelengths a Xenics image sensor (Xeva-2.35-320-TE4 320 x 256 pixels, 30 µm pixel pitch) was used.

The beam splitters and image sensors 2, 3 and 4 were only used for alignment, and were removed while taking the measurements.

## 4.2 Reduction of interference effects

Some interference effects were caused by the cover glass of the main image sensor. For $\lambda$ =0.77 µm (PCO edge 3.1) this can be avoided by slightly tilting the sensor until the spatial frequency of the fringes is sufficiently large to be ignored.

For the other wavelengths and image sensors this method does not work, but since the interference fringes are spatially constant, we eliminated the fringes by averaging of spatially shifted images. The image sensor is mounted on a vertically movable stage (Thorlabs MLJ050). Images are taken for different heights of the sensor with a displacement of multiples of the pixel pitch. The image of the slit moves over the sensor area and thus appears at different positions compared to the interference pattern for each image. Image processing is then used to digitally shift the captured image back considering the difference of the sensor

height. This is possible since the accuracy of the stage is high compared to the pixel pitch. Averaging of the shifted images results in a significant reduction of the interference effect.

## 4.3 Measurement Procedure

Measurements were performed for the six devices mentioned previously (two vendors supplied devices with depths of 0.5mm, 1.0mm, and 1.5mm) for different input and output polarizer settings (S/P) for different positions of the device with respect to

the input knife-edge illumination (in steps of 2.5 µm). For each setting 20 images were recorded: Ten for the illuminated device and ten without illumination as a "dark reference" (to be subtracted during image processing). Figure 8 shows a typical result for four different measurements (only regions near the slit image are shown) and the normalized intensity cross section along the vertical.

## 4.4 Modification for throughput measurement

The throughput was measured with a slightly modified configuration for simplicity. The pickoff mirror was replaced by a circular aperture to illuminate only a defined area. Images were taken under ACT and ALT input polarization. Reference images were taken with thin slits (Edmund #58-541 and #58-542) of known width (measured using a confocal microscope), as well as without air slit or slit homogenizer to account for potential laser intensity fluctuations. The slit homogenizer throughput values are treated as relative to the air slit measurements as the design trade for GeoCarb is between these two options.



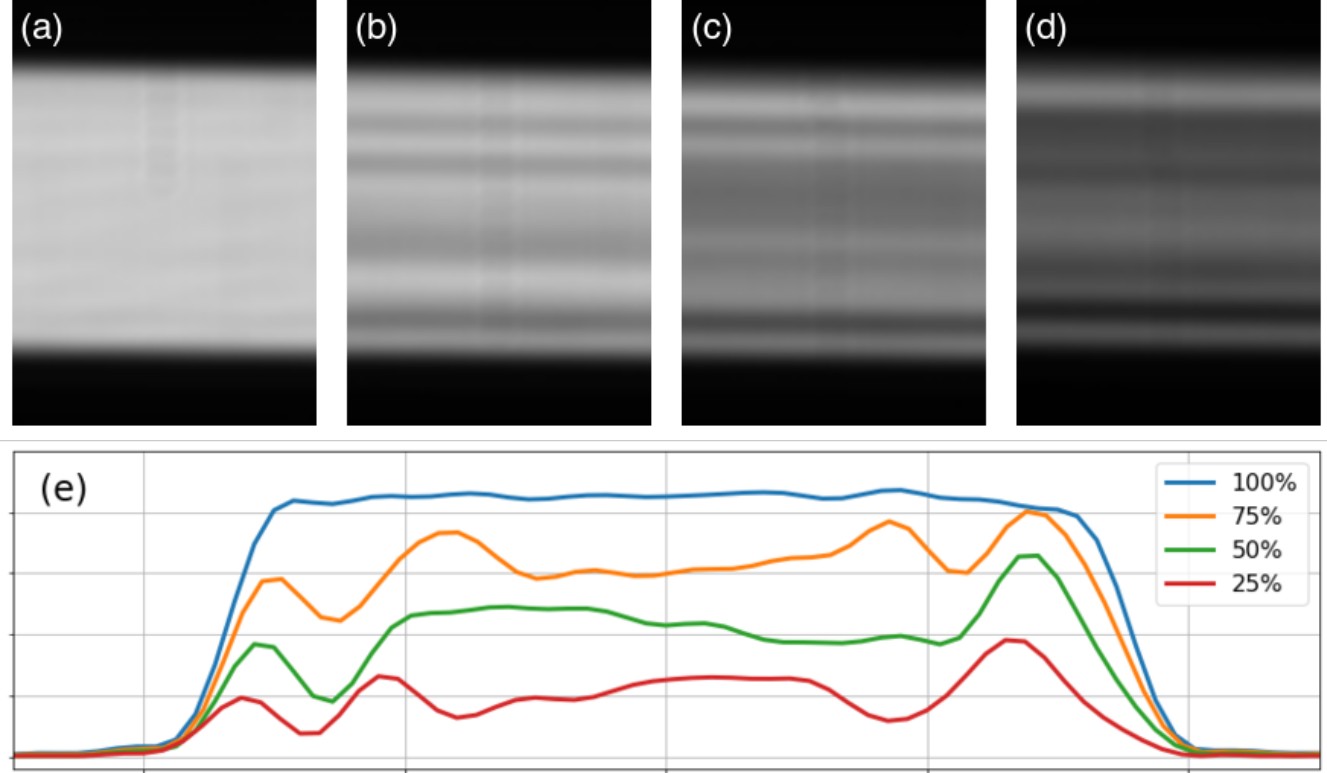

**Figure 8.** (Top) Typical raw images recorded on the image sensor with different levels of illumination controlled by the position of the knife edge (only regions of interest shown): (a) 100%, (b) 75%, (c) 50%, and (d) 25%. (Bottom) 1-D profiles obtained by averaging the panels in the top row in the ACT direction (i.e. in the horizontal direction). Note the decreasing intensity as the knife edge obscures more of the light, apparent in the change in brightness of the figures from left to right.

## 5   Results

The performance test results and model simulations are presented in this section, as well as some further investigations with the model to explain the some unexpected findings uncovered during testing.

### 5.1   1-D SH Measurements

The measurements were taken as described in the previous sections for different positions of the knife edge, and the resulting 2-D images were averaged to produce the 1-D profile of the homogenized slit images. We compare the results for fully illuminated and 50% illuminated test data, as well as the throughput performance below.





### 5.1.1 Homogenization Performance

As described above, 2-D images were taken for each SH as it was moved across the knife edge, gradually obscuring larger and larger portions of the SH. The efficacy of the SH is judged by how well it reproduces the normalized profile of illumination of the fully illuminated SH when the input illumination is partially obscured by the knife edge after averaging in the ACT direction. Importantly, the image of a fully illuminated SH is not the same as that of a typical air slit due to the interference effects. However, these effects are measured and would be taken into account during the pre-flight ISRF characterization (e.g., Lee et al. (2017)) for GeoCarb, and so are not cause for concern. Thus, the homogenization performance for each device is compared against the image of the fully illuminated SH for that device only.

Figure 9 shows the results of the measurements for each of the six devices at the four wavelengths of interest, when the slit is fully illuminated and partially illuminated, for H (left column) and V (right column) polarizations. The 50% illumination results are not normalized in order to show the effects of the reduction of throughput due to partial obscuration of the slit entrance. A few observations are noteworthy. All of the devices reduce the nonuniformity that would be apparent in the 50% illuminated images in the presence of an air slit only, which is apparent from the somewhat more even distribution of light across the width of the slit image. Interference effects due to the multibeam interferences that takes place manifest as oscillations in the 1-D profiles. These effects become stronger the less of the entrance slit is illuminated and weaker as the wavelength grows. We found, somewhat unexpectedly, that the SH devices prefer one polarization (H) over the other (V). This is most apparent as a drop in throughput for V relative to H as seen in Table 2. We note that this effect generally gets more pronounced as the depth of the homogenizer grows, as well as for longer wavelengths, though not for all devices and wavelengths.

### 5.1.2 Relative Throughput measurements

The relative throughput measurements show a strong reduction for vertical polarized light, especially at increased device depth (see table 2). The results show a dependence on vendor, wavelength, and slit homogenizer depth. The narrowing in the V polarization relative to the H polarization increases with depth as well as wavelength (with the exception of the $1.5\,\mathrm{mm}$ device by Vendor 1). These variations imply a preference for incoming light with H polarization, which could complicate interpretation of radiances taken from space where the polarization state in incident light will be unknown. The GeoCarb instrument does not include mitigations for polarization of incoming light, such as a linear polarizer or polarization scrambler.

### 5.2 Comparison of SH Models with Zemax

In order to explain the polarization-dependent features in the SH measurement data, we employ the diffraction SH model. To validate our model, a comparison with simulations using the Zemax software package was performed. A 1-D SH was entered in Zemax non-sequential mode. Two parallel mirrors were defined, coated with gold (using the Rakic parameterization from Rakić et al. (1998)), with a length of 36 $\mu$m and with a length of 1.0 mm. A small source was defined, emitting coherent light at a wavelength of 1600 nm in a cone with 0° angle in the along-slit direction, and a F-number of 4.5 in the across-slit direction.

**Figure 9.** Across-slit illumination as a function of slit position for horizontally (left column) and vertically (right column) polarized light. The colors represent the different devices D1-D6 detailed in Table 2. Solid lines depict fully illuminated scenes, normalized to the maximum value, while dashed lines represent the illumination pattern resulting from 50% of the illumination being blocked by a knife edge (normalized by the maximum value of the corresponding fully illuminated slit image), which is apparent from the lower signal in the dashed lines.





| Device | Manufacturer | Depth | 0.76 µm H | 0.76 µm V | 0.76 µm V vs H (%) |
|---|---|---|---|---|---|
| D1 | Vendor 1 | 0.40 mm | 0.89 | 0.72 | -19.1 |
| D2 | Vendor 2 | 0.46 mm | 0.89 | 0.64 | -28.0 |
| D3 | Vendor 1 | 1.0 mm | 0.84 | 0.57 | -32.1 |
| D4 | Vendor 2 | 0.95 mm | 0.85 | 0.56 | -34.1 |
| D5 | Vendor 1 | 1.50 mm | 0.83 | 0.46 | -44.6 |
| D6 | Vendor 2 | 1.4 mm | 0.80 | 0.48 | -40.0 |
| Device | Manufacturer | Depth | 1.62 µm H | 1.62 µm V | 1.62 µm V vs H (%) |
| D1 | Vendor 1 | 0.40 mm | 0.93 | 0.71 | -23.7 |
| D2 | Vendor 2 | 0.46 mm | 0.89 | 0.59 | -33.7 |
| D3 | Vendor 1 | 1.0 mm | 0.90 | 0.47 | -47.8 |
| D4 | Vendor 2 | 0.95 mm | 0.83 | 0.46 | -44.6 |
| D5 | Vendor 1 | 1.50 mm | 0.80 | 0.43 | -46.3 |
| D6 | Vendor 2 | 1.4 mm | 0.82 | 0.41 | -50.0 |
| Device | Manufacturer | Depth | 2.05 µm H | 2.05 µm V | 2.05 µm V vs H (%) |
| D1 | Vendor 1 | 0.40 mm | 0.92 | 0.75 | -18.5 |
| D2 | Vendor 2 | 0.46 mm | 0.97 | 0.68 | -29.9 |
| D3 | Vendor 1 | 1.0 mm | 0.90 | 0.61 | -32.2 |
| D4 | Vendor 2 | 0.95 mm | 0.94 | 0.60 | -36.2 |
| D5 | Vendor 1 | 1.50 mm | 0.99 | 0.51 | -48.5 |
| D6 | Vendor 2 | 1.4 mm | 0.92 | 0.56 | -39.1 |
| Device | Manufacturer | Depth | 2.33 µm H | 2.33 µm V | 2.33 µm V vs H (%) |
| D1 | Vendor 1 | 0.40 mm | 1.0 | 0.75 | -25.0 |
| D2 | Vendor 2 | 0.46 mm | 0.98 | 0.67 | -31.6 |
| D3 | Vendor 1 | 1.0 mm | 1.01 | 0.52 | -48.5 |
| D4 | Vendor 2 | 0.95 mm | 0.91 | 0.52 | -42.9 |
| D5 | Vendor 1 | 1.50 mm | 0.92 | 0.50 | -45.7 |
| D6 | Vendor 2 | 1.4 mm | 0.96 | 0.50 | -47.9 |

**Table 2.** Throughput as compared to a thin slit for different devices with an F-number of f/4.5 for horizontal (H) and vertical (V) polarization.

More elaborate models could be set up in Zemax with a source emitting a 2-D cone of light, but this would require a more complicated design to create an astigmatic beam, which was not done in this work. Monte-Carlo simulations were used with $10^6$ to $10^7$ rays. Execution time was typically between 1 and 20 minutes depending on the required sampling and targeted accuracy (both impacting the required number of rays).

The SH analytical model assumes a scalar wave, in the sense that optical path differences and phase shifts are calculated for each ray and combined directly to compute the intensity where the rays interfere. By contrast, Zemax propagates the real





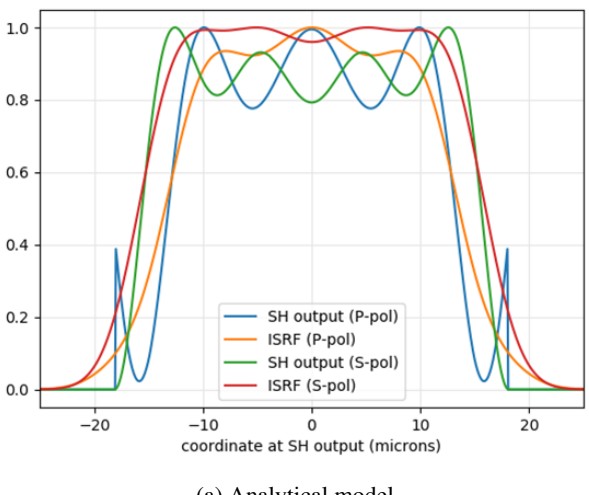

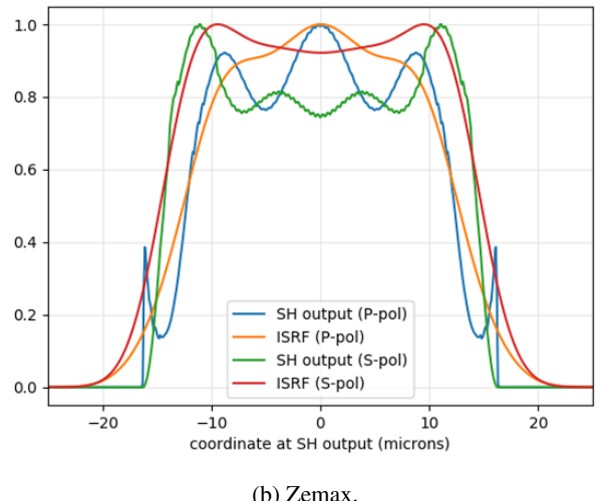

(a) Analytical model.

(b) Zemax.

**Figure 10.** SH output illumination, and resulting ISRF in P and S polarization for a SH length of 1mm, F-number = 4.5 and $\lambda$ = 1600 nm. (a) analytical model, (b) Zemax. A square pupil and an interference model were used.

electric field along the ray and computes interferences with real electric field vectors: interferences are computed separately for the X, Y and Z components, and intensities added at the end.

We found a very good agreement between the Zemax and analytical model results. In Figure 10 we present the obtained intensity profiles at slit output as well as the ISRFs. The ISRFs are computed by convolving the intensity profiles with a gaussian PSF of FWHM equal to 1/6 of the SH width, which represents the optical point spread function of the imaging optics of the experimental setup described in the previous sections. Both models assume a SH width w = 36 $\mu$m, depth L = 1 mm, F-number = 4.5 and $\lambda$ = 1.6 $\mu$m.

### 5.3 Polarization-dependent ISRF Width

We investigated the cause of the polarization-dependence in the simulated and measured ISRFs using the interference SH model. This effect was linked to the different phase shifts gained by the electric field at the reflection on the gold layer. When both dephasings are artificially set to the same value in the model, simulations show nearly identical transfer functions in P- and S-polarization and nearly no variation of the ISRF width, which implies that the difference in reflectance amplitude with polarization plays almost no role.

To understand the role of the phase shift, it is necessary to look at its variations with angle of incidence. In Figure 11, we represent the phase shifts in both polarizations, including the 180° phase offset specific to grazing incidence in P-polarization. When the angle of incidence is very large, or equivalently when the SH is operated with a beam with large F-number, then both phase shifts are close to 180° and there is almost no polarization effect. When the angles of incidence are decreased (i.e.



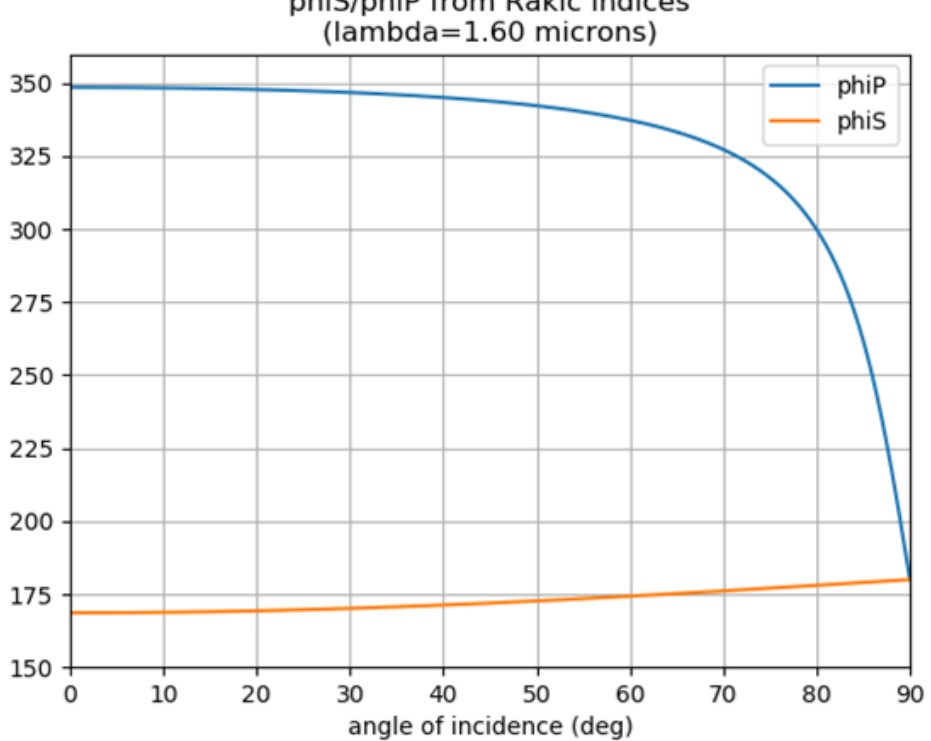

**Figure 11.** P-polarized and S-polarized phase shifts at reflection on gold as a function of the angle of incidence, for $\lambda = 1.6\ \mu$m. The 180 deg phase shift specific to P-pol shown on Figure 5 is included.

going to lower F-numbers) the phase shifts start to differ. In particular, the phase shift in P-polarization starts to become large and easily reaches 270° (so a difference of 90°). Then a clear polarization effect occurs.

This applies for the relatively low F-number of GeoCarb. Additionally, as can be seen on the plots showing the output illumination and the ISRFs, the magnitude of the ISRF width variation is comparable to one oscillation period in the interference pattern. This suggests that the relative change of ISRF width will be smaller for a slit that is significantly wider than the size of one diffraction pattern (Airy disk). For this reason, the polarization effect becomes clearly visible due to GeoCarb's narrow slit (36 microns) and becomes more pronounced at larger wavelengths.

### 5.4  Predicted transmittance

Next, the SH transmittance was investigated. The amount of intensity transmitted by the SH when its input is illuminated with a uniform intensity profile was computed using a diffraction model. Diffraction models are not only more accurate than interference models; they also rigorously conserve energy thanks to the properties of diffraction integrals so they are well





| Wavelength ($\mu$m) | Ratio between Tp and Ts |
|---|---|
| 0.77 | 0.866 |
| 1.62 | 0.613 |
| 2.05 | 0.474 |
| 2.33 | 0.383 |

**Table 3.** Ratio between P-polarized and S-polarized transmittances predicted for GeoCarb (w = 36 $\mu$m, L = 1 mm), using a SH diffraction model and assuming a rectangular pupil (F-number = 4.5).

suited for transmittance calculations. The SH transmittance was obtained with a summation of the transfer matrix along one dimension, which corresponds to the case where the SH input is fully and uniformly illuminated.

The results are presented in Figure 12. Three cases are studied:

– Ts/Tp radiometry only: As a reference, we made a simple SH transmittance calculation with a modified interference model, where all effects from the interference were suppressed. This model gives the net effect of the finite reflectivity of the gold coating, combined with the angles of propagation inside the SH.

    – Ts/Tp diffraction model phase = 180°: the transmittance was computed with the diffraction model, assuming that the phase shift at reflection keeps the ideal value of 180° for all angles of incidence.

– Ts/Tp diffraction model gold: finally the transmittance was computed with the diffraction model and a real gold layer.

We immediately see that the three models predict a nearly identical transmittance in S-polarization that is close to 100%, while they give very different losses in P-polarization. Taking the simulation without interferences (1st curve) as a reference, we notice that including diffraction effects with a 180° phase shift results in a slight increase of the losses. This increase can be easily explained by the fact that diffraction sends some light at larger angles than what geometric optics predicts, creating stronger losses. By contrast, the diffraction model with gold dephasings predicts an even stronger increase of losses. This increase is surprising as it is purely due to the phase shifts at reflection (the amplitude of reflection coefficients being unchanged). The effect is not fully understood, but is consistent with the obtained experimental results shown in Table 2. One possible explanation is that the gold phase shift in P-polarization has a strong influence on the diffracted angles, so that part of the incident light experiences many more reflections inside the SH and a much higher absorption.

## 6   Discussion

The measurements and modeling together provide a strong argument that 1-D slit homogenizers of the type considered for GeoCarb (i.e. plane parallel, gold-coated mirrors) are highly sensitive to the polarization of incoming light. The polarization response of the instrument itself can be characterized in pre-flight calibration, but the polarization state of the incident light is itself unknown. This could lead to significant errors in trace gas retrievals using spectra through the difficulty in disambiguation

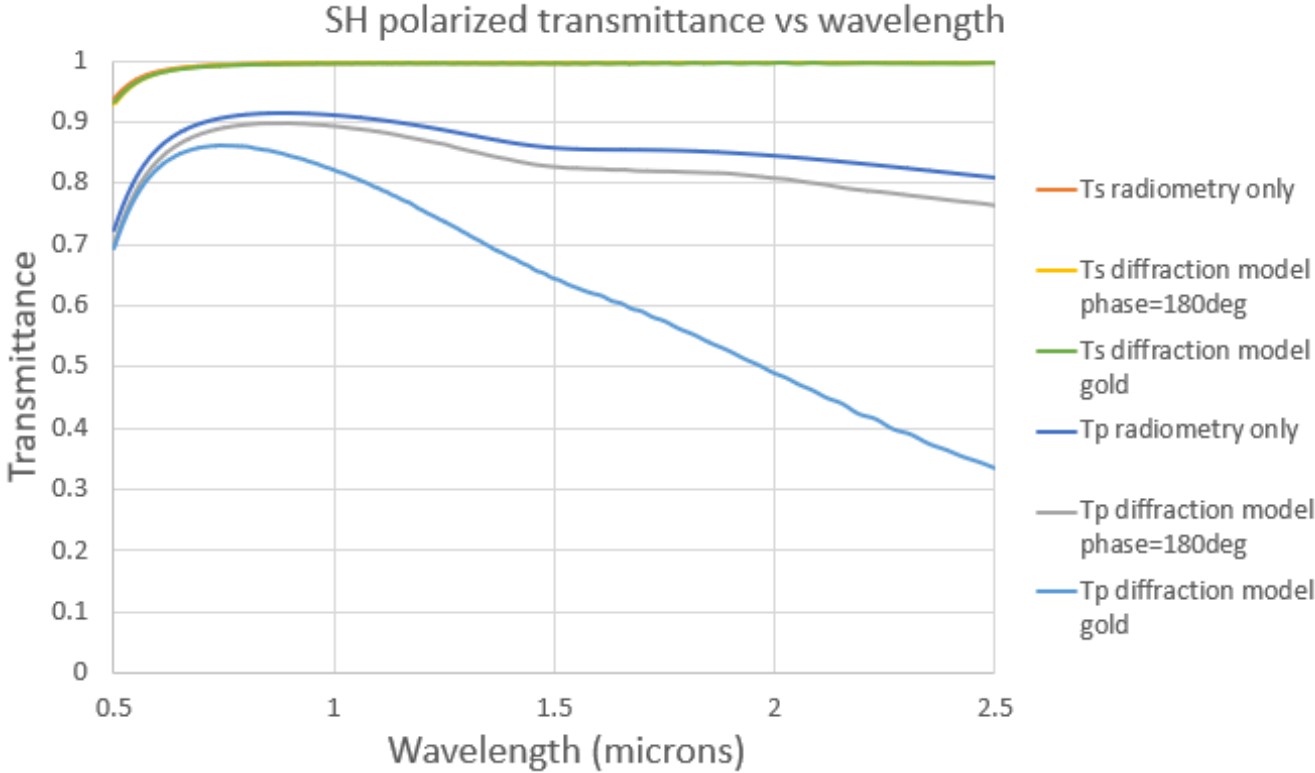

**Figure 12.** SH transmittance vs polarization and wavelength for w = 36 $\mu$m, L = 1 mm and F-number = 4.5

of albedo and polarization states. For example, two scenes with identical trace gas column mole fractions but different polarization states would be hard to distinguish from scenes with identical polarization states but different trace gas amounts, even if the instrument were perfectly characterized. Another important challenge is the implication that the ISRF will be strongly polarization dependent and thus change from scene to scene in a similar way as it would with no slit homogenizer present due to the scene brightness variations. Further, there is the potential of using other space-based sensors to try to get a handle on sur-

face brightness, but no such information exists for polarization. Finally, the signal-to-noise ratio will also be highly dependent on polarization and the lower efficiency of the SH would almost certainly lead to poorer quality retrievals. In a very real sense, the 1-D devices add complexity to the instrument design but do not reduce uncertainty. For this reason, the slit homogenizer was removed from the GeoCarb design.

  The upcoming Sentinel 5 mission includes a 1-D slit homogenizer in their design, as is discussed in Köhler et al. (2021)

and Hummel et al. (2021). The results in this paper do not invalidate the design of the Sentinel 5 instrument because 1) the instrument includes a polarization scrambler and 2) their slit is $120 \mu m$ or 3-4 times wider than GeoCarb, and their F# is close to 10, which is a very different optical regime than the GeoCarb instrument. Future missions that choose to include a slit





homogenizer would do well to perform an optical performance analysis of their slit assembly prior to installation and definitely include an optical element that removes polarization on the incoming light.

**7 Conclusions**

We have presented both experimental and simulated results related to the performance of 1-D SH devices consistent with the GeoCarb optical design. We found that the devices exhibited a strong sensitivity to the polarization of the light incident on the slit. This was apparent in both the width of the slit image as well as the relative throughput of the different devices. The difference between the two orthogonal polarization states grew worse as wavelengths get longer. We then augmented a model

of the slit homogenizer to demonstrate that this narrowing is caused by the reflection on the gold coatings on the mirrors that lead to a phase shift between the P- and S-polarizations. It seems likely that different materials could be used with better results, but the space application required for GeoCarb limited our interest in that investigation.

Future work involves the implementation of these measurement results in a retrieval framework similar to that described in O'Brien et al. (2016) to better understand the tradeoff between the albedo effects and the polarization sensitivities in the two

designs.

*Author contributions.* SC drafted the manuscript and analyzed the measurements and model simulations. TH and MT collected the measurements. JC developed the polarization-dependent SH model. EB analyzed the data. BM and all coauthors provided feedback on the manuscript drafts prior to finalization.

*Competing interests.* The authors declare no competing interests.

*Acknowledgements.* The authors acknowledge funding from NASA through the GeoCarb Mission under award 80LARC17C0001.





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
