# Peer review of "Performance and polarization response of Slit Homogenizers for the GeoCarb Mission"

_Atmospheric Measurement Techniques, 2022_

## Referee Comment (RC2)

**1** General**

The paper is well structured and written. The investigation of effects of polarized scenes on slit homogenizer was missing so far. To our knowledge neither measurements nor simulation exist. In this respect this study is a novelty. While the result on polarization dependence performance is of general interest also the second goal of the study is very interesting, i.e. the dependence on the different devices – even with the same geometry but from different vendors. The result is very helpful for future missions.

Just a remark: The performance of a scene homogenizer for a step and stare system like GeoCarb Mission is more important than for a push broom like Sentinel-5 (due to smearing effects of the moving scene). So it is not understandable why the slit homogenizer has been removed from GeoCarb instead adding a polarization scrambler. Another option would have been fibre-based 2D-slit homogenizer concepts. Maybe this technology was not mature enough by the time of design freeze of GeoCarb, wasn't it? Some comments in this direction would be helpful.

The conclusion from the results seem to be not consistent. In the abstract (line 3f, 9f) one reads that a slit homogenizer is recommended but polarisation should be controlled. While in Discussion you reveal that the SH is not used at all for several technical reason.

Throughout the full paper one gets the impression, that the use of a SH is a mistake even in other missions like Sentinel-5. Only in the last paragraph this impression is corrected. Other missions (like Sentinel-5) may differs from GeoCarb in many aspects, which have a significant impact on the SH performance. As you pointed out these may be different f# at the slit plane or the observation geometry (pushbroom) or the use of a polarization scrambler. The comparison with Sentinel-5 should be neutral and more elaborated. The differences must be given already at the first occurrence of Sentinel-5 in the paper to avoid a misleading understanding.

**2 Detailed comments**

Criticality "major" doesn't mean "not acceptable". It means "would be very nice".

| Text
position         | Critica
lity | Comment                                                                                                                                                                                                                                          |
|--------------------------|-----------------|--------------------------------------------------------------------------------------------------------------------------------------------------------------------------------------------------------------------------------------------------|
| Introduction             | major           | The theme of the paper is on polarisation, therefore the description of GeoCarb should be extended by the information, that it does not include a polarisation scrambling device.                                                                |
| Introduction             | minor           | Why wasn't the usage of fibre homogenizer considered?                                                                                                                                                                                            |
| Introduction             | minor           | Why wasn't a polarization scrambler considered? Were the results from SH too late?                                                                                                                                                               |
| §2.2
formular
L142 | minor           | Please add that "*" means convolution and use different signs for multiplication in line 142, e.g. \cdot like $2 \cdot F \# \cdot w$ or just $2 F \# w$                                                                                          |
| Figure1                  | major           | Please avoid alt and act, and the mixture with north/south, as a geostationary satellite has not a real track direction.                                                                                                                         |
| Figure2                  | minor           | Although it is clear please insert the definition of "slit length" as it is used in the abstract                                                                                                                                                 |
| 3. slit
homogeniser   | minor           | Please provide the f# at slit earlier in the document. The pre-conditioned reader wants to compare the optical conditions of GeoCarb with other instruments (e.g. Sentinel-5), where the paper gives references to.                              |
| 3. slit
homogeniser   | major           | The simulation model is extensively used and an important part of this paper. But it is never described. Please insert the model in form of formula or give a reference where it is described. At least a scratch or changes to existing models. |

| Line 142                              | minor | Please remove or use different signs for multiplication, e.g. \cdot like $2 \cdot F\# \cdot w$ or just $2 \cdot F\# w$                                                                                                                                                       |
|---------------------------------------|-------|------------------------------------------------------------------------------------------------------------------------------------------------------------------------------------------------------------------------------------------------------------------------------|
| Line 148f                             | minor | The following sentence is difficult read: "We applied the computed transfer function for each depth and the resultant coefficient of variation measured."                                                                                                                    |
| Line 190                              | major | Could you please provide the impact (in terms of ISRF distortion) by the given tilt?                                                                                                                                                                                         |
| Figure 8                              |       | Please insert axis labelling – at least in the figure below. Pixel information on the above image would be nice.                                                                                                                                                             |
| §5 (e.g.
Table 2 or
line 322ff) | minor | If possible (only if possible) add explanation why different devices with
the same geometry produce different results. One explanation is e.g. that
the two plane parallel mirrors of the SH couldn't be produced 100% plane
and parallel.                          |
| Line 256                              | minor | The sentence is not clear. To the contrary, I expect that the effect grows with growing wavelength (more prominent in SWIR than in NIR).                                                                                                                                     |
| Line 281                              | major | Usually the ISRF knowledge should be in the order of <1% (of the peak) to not cause severe retrieval errors. As model and measurement deviate by much more than 1%, it is not "a very good agreement" (openly spoken, the modelled ISRF cannot be used in a retrieval).      |
|                                       |       | limitations, especially as details of the Zemax algorithm are not known.
But could you please find a weaker formulation instead of "very good"?                                                                                                                           |
| Figure 11                             | minor | Some axis titles would be nice (and mandatory). Please use the capability of the plotting tool to display Greek letters, to match the parameters names with the descriptive text.                                                                                            |
| around
line300                     | major | Usually spectrometers are polarisation sensitive due to the grating. This may be mentioned here. Further, the issue on ISRF is because the incident radiance is polarised, that could be also mentioned somewhere in the paper. Either here, or in the introduction chapter. |
| Line 322                              | minor | The results are too different to be described as consistent                                                                                                                                                                                                                  |
| Figure 12                             | minor | Orange and yellow are hidden by the green curve. This could be mentioned in the figure caption                                                                                                                                                                               |

---

## Author Response (AR1)

**Reviewer 1**

<General Comments>
Imaging spectrometer with high spectral resolution is a key technology for greenhouse gases and their related species monitoring by remote sensing. Acquired image provides large emission sources and plume information. Very narrow spectral width needs moderate optical throughput to achieve high signal to noise ratio. Inhomogeneity within the footprint distorts the instrument spectral response function (ISRF). Gratings are conventionally used, but they have high polarization. Measurement techniques acquiring solar lights reflected by the Earth's surface and scattered by thin cloud and aerosols should care input light polarization, too. The topics discussed here is challenging and important. My question is which is more critical for GeoCarb: polarization sensitivity or spatial inhomogeneity? Both OCO-2 and GeoCarb are using high resolution imaging spectrometer technology, but they have different swath, spectral coverage, and throughput, The OCO-2 and OCO-3 on orbit already achieved low bias and reduce random errors in $CO_2$ retrieval without using state-of-art slit homogenizer. Are there a critical angle and/or spectral band, and footprint size, where the distortion in ISRF becomes critical for CO2 retrieval? What is the main difference between OCO-2 and GeoCarb?

Added text: Unlike GeoCarb, the Orbiting Carbon Observatories 2 and 3 (OCO-2/OCO-3) were designed with a slight defocus in order to reduce the effects of within scene brightness variations.  This choice, together with much smaller spatial footprints and averaging along track in Low Earth Orbit, are likely to make this concern less problematic for OCO-2 and OCO-3 than for GeoCarb, though no conclusive studies have been performed (David Crisp, personal communication).

Slit design, method, and results of characterization test in the laboratory are well described. I recommend minor revision before publication.

<Specific Comments>
(1) Page 4. MODIS data
If it is a real data, the location and date of the observation should be described.

Good point.  We added the correct date and location to the figure.  For reference, this is a GeoCarb footprint centered on the Southern Great Plains DOE ARM site as seen from MODIS on 6/18/2014 from the 8-day 500m black sky albedo product.

(2) Page 10, Fig. 6 The right figure

Is "analyzer" a "polarizer" in the middle figure? Description of analyzer is needed.

Yes, the analyzer is a polarizer identical to the one prior to the slit homogenizer. This is the typical parlance of the field for the polarizer following the device. We have updated the language.

(3) Page 10, Line 187 "noise"
Is it random electrical noise?

We added "shot and read noise arising from the light source and image sensors" to clarify this in the text.

(4) Page 13, Figure 8, "different level of illumination"
More detailed explanation such as definition of "level", will help readers' understanding.

We rewrote this to be more descriptive. "(Top) Typical raw images recorded on the image sensor as the knife edge obscures more and more of the incoming beam (only regions of interest shown): (a) Fully Illuminated (100\%) (b) 75\% unobstructed, (c) 50\% unobstructed, and (d) 25\% unobstructed. (Bottom) 1-D profiles obtained by averaging the panels in the top row along vertical cross sections in the top image, which would correspond to variations across the shorter dimension of the slit. Note the decreasing intensity as the knife edge obscures more of the light, apparent in the change in brightness of the figures from left to right. Note also that the slit homogenizer effectively reduces the sharp gradient that would be present along a knife edge with a typical slit and spreads the light fairly uniformly across the image."

Reviewer 2

**General**

The paper is well structured and written. The investigation of effects of polarized scenes on slit homogenizer was missing so far. To our knowledge neither measurements nor simulation exist. In this respect this study is a novelty. While the result on polarization dependence performance is of general interest also the second goal of the study is very interesting, i.e. the dependence on the different devices – even with the same geometry but from different vendors. The result is very helpful for future missions.

Just a remark: The performance of a scene homogenizer for a step and stare system like GeoCarb Mission is more important than for a push broom like Sentinel-5 (due to smearing effects of the moving scene). So it is not understandable why the slit homogenizer has been removed from GeoCarb instead adding a polarization scrambler. Another option would have been fibre-based 2D-slit homogenizer concepts. Maybe this technology was not mature enough by the time of design freeze of GeoCarb, wasn't it? Some comments in this direction would be helpful.

**These are good points. We were a bit stuck with the design we had. It was much simpler to remove the SH than to add any more optics to the system. There was no real place to put a scrambler or polarizer without a significant redesign. The 2D SH devices were not through environmental testing at the time we were making our design decisions.**

The conclusion from the results seem to be not consistent. In the abstract (line 3f, 9f) one reads that a slit homogenizer is recommended but polarisation should be controlled. While in Discussion you reveal that the SH is not used at all for several technical reason. Throughout the full paper one gets the impression, that the use of a SH is a mistake even in other missions like Sentinel-5. Only in the last paragraph this impression is corrected. Other missions (like Sentinel-5) may differs from GeoCarb in many aspects, which have a significant impact on the SH performance. As you pointed out these may be different f# at the slit plane or the observation geometry (pushbroom) or the use of a polarization scrambler. The comparison with Sentinel-5 should be neutral and more elaborated. The differences must be given already at the first occurrence of Sentinel-5 in the paper to avoid a misleading understanding.

**We understand and agree. We definitely don't want to give the sense that Sentinel 5 made poor design decisions. We added further text describing the difference between LEO and GEO sampling in the introduction:**
**"Simulation studies for the TROPospheric Monitoring Instrument (TROPOMI) \citep{Landgraf2016,Hu2016} have also shown that variations in brightness within a scene, which affect the shape of the instrument spectral response function (ISRF), can lead to errors in NIR and SWIR trace gas retrievals. Other internal studies suggested that this problem could be especially challenging for a "step-and-stare" observing platform like GeoCarb. The Orbiting Carbon Observatories 2 and 3 (OCO-2/OCO-3) were designed with a slight defocus in order to reduce the effects of within scene brightness variations. This choice, together with much smaller spatial footprints and averaging along track in Low Earth Orbit (LEO), are likely to make this concern less problematic for OCO-2 and OCO-3 than for GeoCarb, though no conclusive studies have been performed (David Crisp, personal communication). Alternatively, the Sentinel 5 LEO mission, which has comparable spatial resolution to GeoCarb, performed several studies (e.g., \cite{Meister2017}, \cite{Sierk2017}) that deduced that a hardware solution in the form of a "slit homogenizer" (SH) could mitigate these errors successfully in conjunction with along-track averaging."**

**Detailed comments**

Criticality "major" doesn't mean "not acceptable". It means "would be very nice".

| Text position | Criticality | Comment |
|---|---|---|
| Introduction | major | The theme of the paper is on polarisation, therefore the description of GeoCarb should be extended by the information, that it does not include a polarisation scrambling device. |
| Introduction | minor | Why wasn't the usage of fibre homogenizer considered? |
| Introduction | minor | Why wasn't a polarization scrambler considered? Were the results from SH too late? |
| §2.2 formular L142 | minor | Please add that "*" means convolution and use different signs for multiplication in line 142, e.g. \cdot like $2 \cdot F\# \cdot w$ or just $2\ F\#\ w$ |
| Figure1 | major | Please avoid alt and act, and the mixture with north/south, as a geostationary satellite has not a real track direction. |
| Figure2 | minor | Although it is clear please insert the definition of "slit length" as it is used in the abstract |
| 3. slit homogeniser | minor | Please provide the f# at slit earlier in the document. The pre-conditioned reader wants to compare the optical conditions of GeoCarb with other instruments (e.g. Sentinel-5), where the paper gives references to. |
| 3. slit homogeniser | major | The simulation model is extensively used and an important part of this paper. But it is never described. Please insert the model in form of formula or give a reference where it is described. At least a scratch or changes to existing models. |

| Line 142 | minor | Please remove or use different signs for multiplication, e.g. \cdot like $2 \cdot F\# \cdot w$ or just $2\ F\#\ w$ |
|---|---|---|
| Line 148f | minor | The following sentence is difficult read: "We applied the computed transfer function for each depth and the resultant coefficient of variation measured." |
| Line 190 | major | Could you please provide the impact (in terms of ISRF distortion) by the given tilt? |
| Figure 8 | | Please insert axis labelling – at least in the figure below. Pixel information on the above image would be nice. |
| §5 (e.g. Table 2 or line 322ff) | minor | If possible (only if possible) add explanation why different devices with the same geometry produce different results. One explanation is e.g. that the two plane parallel mirrors of the SH couldn't be produced 100% plane and parallel. |
| Line 256 | minor | The sentence is not clear. To the contrary, I expect that the effect grows with growing wavelength (more prominent in SWIR than in NIR). |
| Line 281 | major | Usually the ISRF knowledge should be in the order of <1% (of the peak) to not cause severe retrieval errors. As model and measurement deviate by much more than 1%, it is not "a very good agreement" (openly spoken, the modelled ISRF cannot be used in a retrieval).

I appreciate the quality of the comparison and I understand that there are limitations, especially as details of the Zemax algorithm are not known. But could you please find a weaker formulation instead of "very good"? |
| Figure 11 | minor | Some axis titles would be nice (and mandatory). Please use the capability of the plotting tool to display Greek letters, to match the parameters names with the descriptive text. |
| around line300 | major | Usually spectrometers are polarisation sensitive due to the grating. This may be mentioned here. Further, the issue on ISRF is because the incident radiance is polarised, that could be also mentioned somewhere in the paper. Either here, or in the introduction chapter. |
| Line 322 | minor | The results are too different to be described as consistent |
| Figure 12 | minor | Orange and yellow are hidden by the green curve. This could be mentioned in the figure caption |

Introduction: The theme of the paper is on polarisation, therefore the description of GeoCarb should be extended by the information, that it does not include a polarisation scrambling device.

Added this to the description of the GeoCarb instrument in Section 2.1.

Introduction: Why wasn't the usage of fibre homogenizer considered?

It was considered, but that device was not of appropriate technical readiness level at the time of our PDR. Further environmental testing would be required that was not within the scope of the mission schedule and budget.

Introduction: Why wasn't a polarization scrambler considered? Were the results from SH too late?

The GeoCarb design was considered frozen once the 1D SH was added. There was no good option for placement of any polarization mitigation optics. Removing the SH required very small changes, while adding any optics would have required a major redesign.

§2.2 formular L142: Please add that "*" means convolution and use different signs for multiplication in line 142, e.g. \cdot like $2 \cdot F\# \cdot w$ or just $2\,F\#\,w$

Thanks for catching this. Fixed!

Figure 1: Please avoid alt and act, and the mixture with north/south, as a geostationary satellite has not a real track direction.

We have corrected this and replaced ALT and ACT as appropriate in this figure and the text.

Figure 2: Although it is clear please insert the definition of "slit length" as it is used in the abstract

Done. We added the reference to the X dimension in the axes below.

Section 3: Please provide the f# at slit earlier in the document. The pre-conditioned reader wants to compare the optical conditions of GeoCarb with other instruments (e.g. Sentinel-5), where the paper gives references to.

We added information related to the slit and f# in Section 2.1.

Section 3: The simulation model is extensively used and an important part of this paper. But it is never described. Please insert the model in form of formula or give a reference where it is described. At least a scratch or changes to existing models.

This is an important reference that is missing. We have added the reference to a specific equation from Bauer et al (2017) in Section 3. There is a significant amount of writing that has already been done on this topic that we hoped to avoid repeating.

Line 142: Please remove or use different signs for multiplication, e.g. \cdot like $2 \cdot F\# \cdot w$ or just $2\,F\#\,w$

Done, thanks.

Line 148: The following sentence is difficult read: "We applied the computed transfer function for each depth and the resultant coefficient of variation measured."

We replaced that paragraph with the following text:

"To determine the optimum depth for the slit homogenizer for GeoCarb, we simulated polarization-independent transfer matrices for depths from 0.1\unit{milli}{meter} to 2\unit{milli}{meter} at all four wavelengths of interest (listed in Table 1) using the previously documented SH model

\citep{Meister2017}. We then randomly generated an ensemble of input scenes with inhomogeneities that had an average coefficient of variation of scene brightness of 20\% on size scales of about 1/5th of a slit width. We applied the computed transfer matrix for each depth  to the ensemble and the resultant coefficient of variation of the output was calculated. The median reduction in coefficient of variation for each wavelength at each slit depth was used as the metric for selection."

Line 190: Could you please provide the impact (in terms of ISRF distortion) by the given tilt?

This should not affect the results because: 1) The tilt was in the along-slit direction, so it should not affect the ISRF where variations are important in the across-slit direction; 2) Any effect would be a scaling by the cosine of the angle of tilt, which was very small.  We added some text to this effect at this location.

Figure 8: Please insert axis labelling – at least in the figure below. Pixel information on the above image would be nice.

The axes are normalized to the maximum of the 100% illumination image.  We modified the image with axis labels as well as the caption to make things clearer.

§5 (e.g. T able 2 or line 322ff): If possible (only if possible) add explanation why different devices with the same geometry produce different results. One explanation is e.g. that the two plane parallel mirrors of the SH couldn't be produced 100% plane and parallel.

Yes, as with most things, workmanship will vary from optic to optic for prototypes.  The two vendors had different coating processes that yielded different surface roughnesses, for example.  The degree to which the SH devices were plane parallel was likely also a factor.

Line 256: The sentence is not clear. To the contrary, I expect that the effect grows with growing wavelength (more prominent in SWIR than in NIR).

This is what was meant. We changed the wording to make it clearer.

Line 281: Usually the ISRF knowledge should be in the order of <1% (of the peak) to not cause severe retrieval errors. As model and measurement deviate by much more than 1%, it is not "a very good agreement" (openly spoken, the modelled ISRF cannot be used in a retrieval). I appreciate the quality of the comparison and I understand that there are limitations, especially as details of the Zemax algorithm are not known. But could you please find a weaker formulation instead of "very good"?

We softened this language.  The point of the simulations was to provide a plausible explanation of the behavior we were seeing in the measurements.  The qualitative agreement we see with Zemax suggests that we can use the analytical model to investigate the sensitivity to surface coatings, for example.

Figure 11: Some axis titles would be nice (and mandatory). Please use the capability of the plotting tool to display Greek letters, to match the parameters names with the descriptive text.

We updated the figure to be clearer with the axis titles and better legend.

Line 300: Usually spectrometers are polarisation sensitive due to the grating. This may be mentioned here. Further, the issue on ISRF is because the incident radiance is polarised, that could be also mentioned somewhere in the paper. Either here, or in the introduction chapter.

Added a bit more in the discussion section to make sure it's clear that the incoming light is polarized.

Line 322: The results are too different to be described as consistent

We softened this language – agree that the results are mixed in terms of their agreement.

Figure 12: Orange and yellow are hidden by the green curve. This could be mentioned in the figure caption

Done.